# Heterogenization of a Tungstosilicic Acid Catalyst for Esterification of Bio-Oil Model Compound

Prapaporn Prasertpong [1], Jeremiah Lipp [2], Anhua Dong [2], Nakorn Tippayawong [3],* and John R. Regalbuto [2],*

1    Department of Mechanical Engineering, Rajamangala University of Technology Thanyaburi,
     Pathum Thani 12110, Thailand
2    Department of Chemical Engineering, University of South Carolina, Columbia, SC 29208, USA
3    Department of Mechanical Engineering, Faculty of Engineering, Chiang Mai University,
     Chiang Mai 50200, Thailand
*    Correspondence: n.tippayawong@yahoo.com (N.T.); regalbuj@cec.sc.edu (J.R.R.)

**Abstract:** Based on a prior demonstration of the high activity of a homogeneous tungstosilicic acid catalyst for the esterification of acetic acid as bio-oil model compound, a further study has been undertaken in an attempt to heterogenize the catalyst. Tungsten oxide was supported on amorphous silica (W/A150) using incipient wetness impregnation and incorporated into the structure of structured silica (W-KIT-5) via a one-step hydrothermal synthesis. The catalysts were characterized by X-ray diffraction (XRD), X-ray photoelectron spectroscopy (XPS), physisorption (BET), and temperature-programmed desorption of ammonia ($NH_3$-TPD). Both series were evaluated for the esterification of acetic acid with ethanol and compared with the homogeneous 12-tungstosilicic acid catalyst. The result of XRD analysis suggests the average crystallite size of the W oxide nanoparticles on both supports to be less than 2 nm, while XPS analysis revealed that all W existed in the W 6+ oxidation state. From the BET and $NH_3$-TPD analyses, it was shown that the KIT-5 series had higher surface area and acidity than the W/A150 catalyst. The 10% W-KIT-5 was shown to be the best heterogeneous catalyst with the highest activity and acid conversion of about 20% and 93% of the homogeneous catalyst. Significant leaching of tungsten from both the supports occurred and will have to be solved in the future.

**Keywords:** clean energy; fuel upgrading; heteropoly acid; supported tungsta catalysts

## 1. Introduction

Currently, clean renewable and sustainable energy has gained increasing attention due to the continuing concern over fossil fuel depletion and severe environmental problems [1,2]. Biomass is one of the promising alternative fuel sources due to its abundance, with low emission of greenhouse gases or other pollutants [3,4]. Biomass conversion with fast pyrolysis has gained attention in recent years as the conventional method for producing high yields of liquid fuel known as bio-oil [5–7]. Bio-oil has significantly different physical properties from petroleum-derived oils that come from the chemical composition of bio-oil [8]. This composition consists of water and a complex mixture of organic compounds, most of which consist of acids, alcohols, aldehydes, esters, ethers, furans, ketones, phenols, sugars, and oligomers derived from lignin [9–11]. Some of these compounds lead to poor fuel characteristics such as high moisture content, high viscosity, high oxygen content, high corrosiveness, low heating value, and thermal instability—all of which limit the direct use of bio-oil as a transportation fuel [12–15].

Because of these poor properties, bio-oil needs to be upgraded for its subsequent applications. Various upgrading techniques, such as catalytic pyrolysis, catalytic cracking, hydrodeoxygenation, hydrogenation, and steam reforming have been proposed to refine the bio-oil into utilizable liquid fuels [16]. However, bio-oil upgrading through this hydrotreating, and reforming processes still suffers from low yields of upgraded product and

high yields of undesirable products such as char, coke, and tar, which can deposit on the surface of catalysts leading to an increased cost for product recovery [17–19]. Alternatively, esterification has been proposed by several research groups as a simple and effective upgrading technique that can convert acids in the bio-oil to their corresponding esters [20–23], which results in reduced acidity and increased heating value. Furthermore, the reduced acidity improves bio-oil stability since the oligomerization and polymerization reactions are acid catalyzed [24–26].

Mineral acids such as phosphoric acid and sulfuric acid have been regularly and widely used for esterification in industrial applications due to their lower sensitivity toward the presence of free acids and water [27]. However, challenges associated with these acids include equipment corrosion, high yields of byproducts, and environmental concerns [28]. Alternatively, heteropoly acids (HPAs) have gained increasing acceptance as promising alternative catalysts for esterification processes because of their low corrosiveness and low toxicity, as well as very strong Brønsted acidity that can increase the reaction rate under mild conditions [29–32]. For example, high acid conversion via esterification over heteropoly acid catalysts was reported for biodiesel production [32–36] and ethyl acetate synthesis [37]. Nowakowski et al. [38] have reported that bio-oil upgrading via esterification over an HPA catalyst is effective in converting the acids in bio-oil to their corresponding esters by reacting with n-butanol. We have recently reported that a similar process is also effective with ethanol [39,40].

It is generally known that HPA catalysts have a very high solubility in polar solvents such as alcohol and water which are a reactant and products in the esterification process [41]; this is consistent with our observations reported in Prasertpong et al. [40] for the esterification of bio-oil model compounds with ethanol. This can lead to difficulties in separating the HPA catalysts from the reacting mixture. Solid catalysts have also been used for esterification; the main benefit of these catalysts is the ease of separation after the reaction via filtration [42,43]. Moreover, the high catalytic activity of solid catalyst was reported by Embong et al. [44] for biodiesel production via esterification of palm fatty acid distillate catalyzed by rice husk ash $(NiSO_4)/SiO_2$ catalyst with 93% of methyl esters conversion. Tungsten supported by various materials is one of the solid catalysts considered as the catalytically active species for esterification. For example, Oliveira [45] reported that methyl esters conversion of 88% for biodiesel production was acquired via esterification with 12-tungstophosphoric acid supported on zirconia ($H_3PW/ZrO_2$ catalyst). Moreover, the ethyl acetate synthesis by esterification using heteropoly acid supported on montmorillonite K10 catalyst gave acid conversion of 90% and ethyl acetate selectivity of 100%, as reported by Gurav et al. [46]. Specifically, tungsten oxide supported material is one of the well-known catalysts for the acid catalyzed reaction. Mitran et al. [43] found that $WO_3/SiAl$ catalysts showed good selectivity and reusability for n-butyl acetate production through esterification of acetic acid with n-butanol.

Mesoporous silicates are promising support due to their pore structure which facilitates mass transport, high surface area, and pore volume [47]. The large surface area and high total acidity catalyst for biodiesel production were successfully synthesized by Patel [36] using anchoring 12-tungstophosphoric acid to SBA-15 mesoporous silica support. Similarly, Hu et al. [48] found that tungsten supported on SBA-15 mesoporous material provided 92.6% of conversion and 68.3% of selectivity for the conversion of 1-butene to olefins. Chen et al. [49] reported the esterification of oleic acid with methanol catalyzed by 12-silicotungstic acid supported on MCM-41 mesoporous catalyst (HSiW/MCM-41) producing 81.2 wt% yields of methyl oleate. The three-dimensional mesoporous silica support KIT-5 displays interesting properties due to its highly ordered cubic symmetry with tunable cage-type pores [50]. However, the high acidity and performance of the mesoporous silica-supported tungsten catalysts have not yet been widely investigated as the catalysts for bio-oil upgrading via esterification.

In this work, the synthesis and characterization of heterogeneous tungsten oxide catalysts supported on amorphous silica and mesoporous KIT-5 silica were carried out.

In addition, the catalytic activities and effect of the synthesized heterogeneous catalysts on bio-oil model compound upgrading via esterification were investigated and compared with homogeneous tungstosilicic acid. Furthermore, the stability of the synthesized heterogeneous catalysts was examined.

## 2. Results and Discussion

### 2.1. Catalyst Characterization

2.1.1. Structural and W Species Characterization

The X-ray diffraction patterns of the W/A150 and W-KIT-5 samples are shown in Figure 1. The absence of sharp peaks indicates that the coherently diffracting crystallite sizes are very small and makes identification of the crystal phase and estimation of the particle size difficult. The patterns consist mainly of an amorphous silica signal with a broad peak centered around 21.4° (2θ) for the W/A150 samples and 22.4° (2θ) for the W-KIT-5 samples, with small contributions from the W-containing crystallites mainly visible at 26° and 53° (2θ). These contributions clearly increase in intensity in proportion to the W loading.

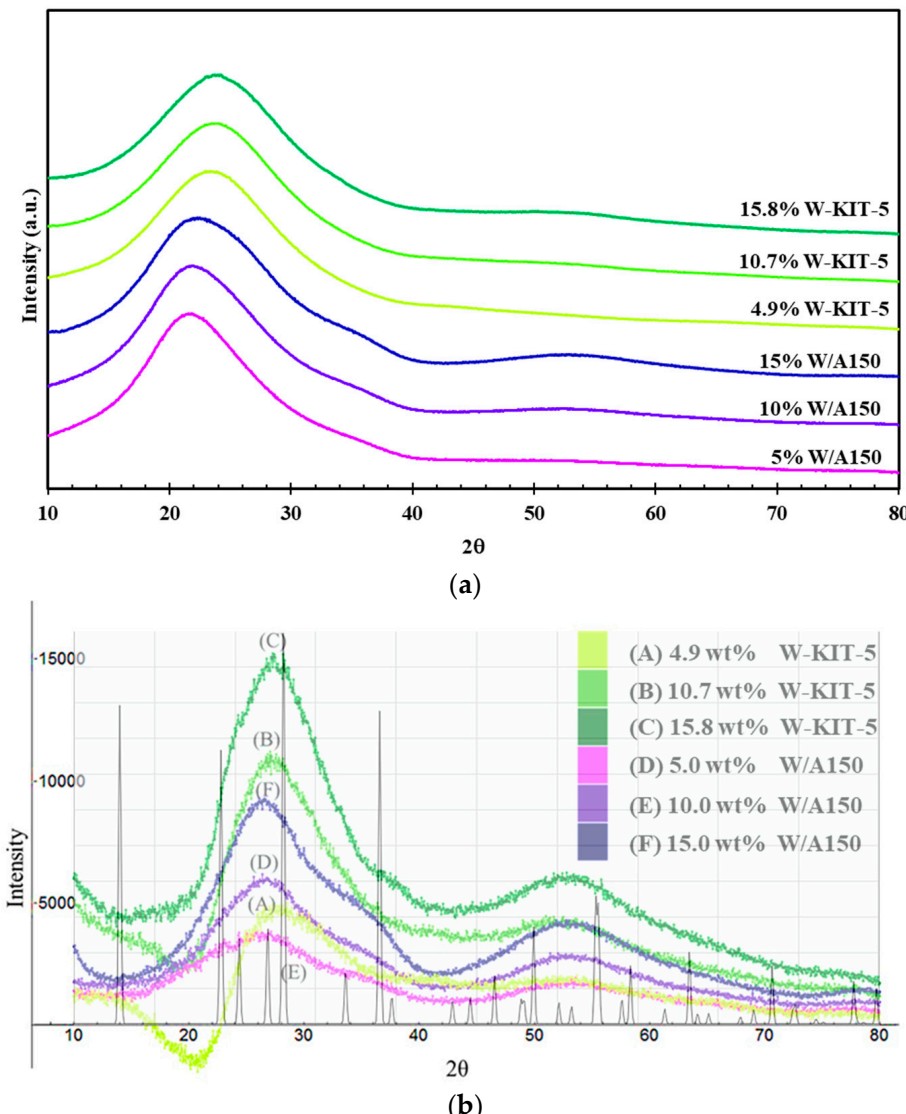

**Figure 1.** X-ray diffraction results of synthesized W/A150 and W-KIT-5 catalysts; (**a**) raw X-ray diffraction patterns; (**b**) background-subtracted patterns.

Further XRD analysis was undertaken for the W/A150 samples, utilizing reference samples to estimate the background, in a method similar to that to be reported by Prasert-pong et al. [51]. The analysis suggests that each of the W/A150 samples consists of crystallites with particle sizes less than 2 nm and that fit best with a hexagonal form of WO$_3$ [52]. More details are given in the supplementary information.

### 2.1.2. Chemical State of W Characterization

The chemical state of tungsten was probed by XPS. The spectra of the W 4f region of all catalysts were recorded and compared for the different metal loadings. The W 4f spectrum of both W/A150 and W-KIT-5 catalysts, displays two main doublet peaks at the binding energies of 36.0 eV and 35.3 eV and in some cases a smaller doublet at 34.4 eV. These W 4f peaks can be attributed to the oxidation states of 6+, 5+ and 4+, respectively, as shown in Figure 2 [53–55]. Plots of the relative amount of W 6+ from the deconvoluted peak areas are shown in Figure 3, where it is seen that the relative amount of W 6+ in the catalysts increases with W loading and is the main component of W/A150 and W-KIT-5 catalysts at higher loadings. It is expected from the literature that the W 6+ oxidation state is desired to achieve higher reactivity, but in some cases, high calcination temperature (~700 °C) is needed, in order to fully create the W 6+ state [56]. In the current study the calcination temperature (300 °C), used to prepare the catalysts yielded a mixture of W oxidation states as expected from similar studies in the past [57]. The O 1s XPS peak deconvolution showed that part of the W has reacted with the support creating W-OH and Si-O-W bonds, responsible for the 5+ and 4+ W oxidation states, present on the catalysts [53]. The W/Si atomic ratio as measured by XPS is shown in Table 1. The increase in the W/Si ratio follows an increase in the W loading for both series but the slightly higher ratios for the A150 series suggest that the W dispersion is slightly higher in that series. The same W/Si ratio between corresponding samples also indicates that the catalytic differences should be explained by the relative amount of W 6+ oxidation state in each sample.

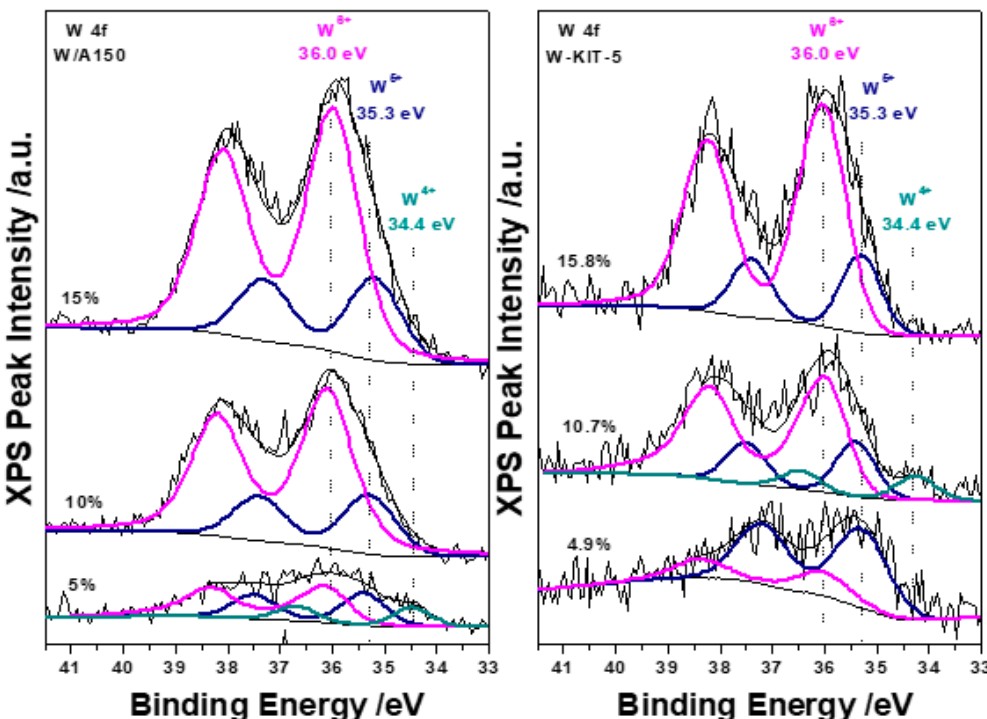

**Figure 2.** XPS spectra of the W 4f photoelectron peak for W/A150 and W-KIT-5, showing the 6+, 5+, and 4+ oxidation states of W that are present on the catalysts.

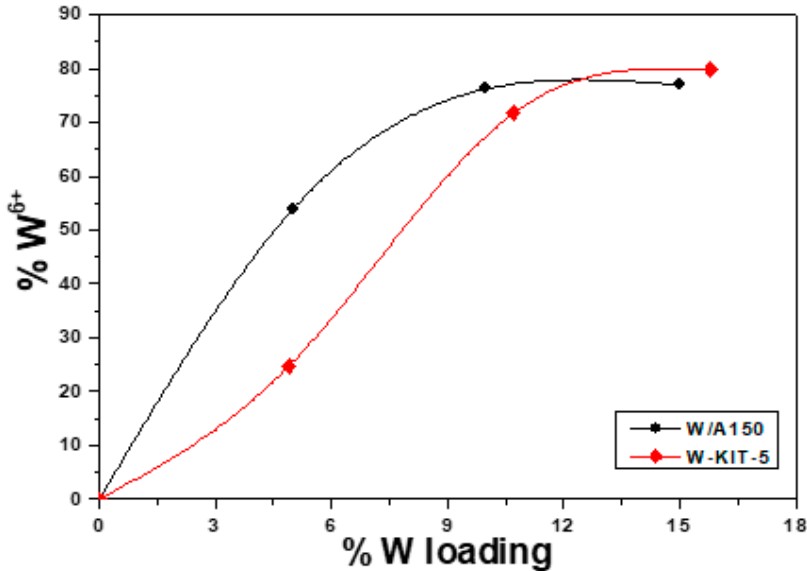

**Figure 3.** The % relative amount of W 6+ oxidation state detected in each catalyst, as a function of the % W loading.

**Table 1.** Chemical characteristics, textural properties, and acidity of the synthesized catalysts.

| Catalyst (wt% of W) | Calculated W/Si (Atomic Ratio) | Surface Area ($m^2/g$) | Pore Volume ($cm^3/g$) | Pore Size (nm) | Total Acidity (mmol $NH_3/g$) |
|---|---|---|---|---|---|
| A150 | - | 153 | 1.21 | 31.6 | 0.6 |
| W/A150 (5%) | 0.004 | 129 | 1.01 | 31.4 | 1.4 |
| W/A150 (10%) | 0.011 | 140 | 1.08 | 30.9 | 2.4 |
| W/A150 (15%) | 0.022 | 113 | 0.79 | 28.1 | 2.4 |
| KIT-5 | - | 673 | 0.40 | 2.4 | 10.5 |
| W-KIT-5 (4.9%) | 0.004 | 795 | 0.55 | 2.7 | 11.6 |
| W-KIT-5 (10.7%) | 0.009 | 743 | 0.52 | 2.8 | 16.5 |
| W-KIT-5 (15.8%) | 0.019 | 702 | 0.54 | 3.1 | 16.0 |

### 2.1.3. Textural Characterization

The properties of the different catalysts are shown in Table 1. W-KIT-5 showed a decrease in surface area and increase in pore size with increasing tungsten oxide content that was consistent with the works reported for tungsten-containing hexagonal mesoporous silica (HMS) [58], tungsten-substituted mesoporous SBA-15 [48], and W-KIT-6 [59]. For W/A150, the pore size was decreased with increasing tungsten oxide content, while the highest surface area was found at 10 wt% tungsten content.

### 2.1.4. Acidity Characterization

The acid strengths of the catalysts were determined by ammonia-TPD analysis and are shown in the last column of Table 2. The acidity of the catalysts increased with increasing tungsten oxide content, which is consistent with that reported for tungsten containing MCM-41 and SBA-15 [60]. The maximum acidity was reached for 10 wt% W/A150 and 15 wt% W/A150. The total acidity of W-KIT-5 was obviously higher than W/A150 indicating the significantly higher acidity of the catalyst prepared with the one-step hydrothermal compared with the catalyst prepared with incipient wetness impregnation.

**Table 2.** Reaction rate of tungstosilicic acid and synthesized catalysts.

| Catalyst (wt% of W) | Reaction Rate (mol/gW/h) |
|---|---|
| Tungstosilicic acid | 0.50 |
| W/A150 (5%) | 0.06 |
| W/A150 (10%) | 0.05 |
| W/A150 (15%) | 0.07 |
| W-KIT-5 (4.9%) | 0.08 |
| W-KIT-5 (10.7%) | 0.07 |
| W-KIT-5 (15.8%) | 0.10 |

*2.2. Catalytic Activity*

The catalytic activities of the catalysts for the esterification of acetic with ethanol are shown in the conversion versus time plots, shown in Figure 4, and the reaction rates of all catalysts are summarized in Table 2. Reaction rates are quantified from the initial slopes of the conversion versus time plots, after the initial lag in reactivity observed in all the supported catalysts, during the diffusion and adsorption of the reactant [61]. It was found that the homogeneous tungstosilicic acid catalyst gave an order of magnitude higher reaction rate than the heterogeneous catalysts. This was possible because the homogeneous catalyst is intimately mixed with the acetic acid in the bulk of the liquid phase, whereas in the heterogeneous catalysts, the reaction rate is limited by mass transfer inside the pores of the catalyst [19]. The W-KIT5 series showed significantly higher activity than the W/A150 series despite the latter being slightly better dispersed and possessing higher pore volume and pore size (well in the mesoporous range) which would improve internal diffusion. The higher activity of the KIT-5 catalysts can therefore be attributed to the higher acidity of the KIT-5 material [26,62].

*2.3. Esterification of Bio-Oil Model Compound*

For bio-oil upgrading via esterification process, acetic acid was used as the bio-oil model compound and ethanol was used as the reactant. The conversion of acetic acid over different catalysts was studied and shown in Table 3. The tungstosilicic acid exhibited the highest catalytic activity with 87% of acetic acid conversion into esters due to the association of the catalyst with acid in the liquid phase. The higher acid conversion of W-KIT-5 series compared with the W/A150 series was observed as their higher acidity and active size which is consistent with the work reported for HPW/SBA-15 catalyst synthesis via impregnation method [63] and direct synthesis route [63,64]. The maximum acid conversion of 81% and 59% was observed over 10 wt% W-KIT5 and 5 wt% W/A150, respectively. However, all two heterogeneously catalyzed reactions were an order of magnitude higher in activity than the reaction without catalyst.

*2.4. Reusability of the Catalyst*

In order to investigate the stability and recyclability of the W/A150 and W-KIT-5 catalysts, additional runs were performed with recovered catalyst as shown in Figure 5. It was observed that both the W-KIT-5 and W/A150 catalysts showed a prominent decrease in conversion with successive cycles. One reason for the deactivation of the catalyst may be the leaching of tungsten from the supporting material, which is shown by ICP analysis in Figure 6. It was found that roughly half of the tungsten leaches from the support into the solution after each experimental run; this leads to a decreasing amount of tungsten in the reaction flask during successive experimental runs. Moreover, it was observed that the leaching of the catalyst was increasing with the catalyst concentration (Figure 6). The catalyst deactivation through leaching of the catalyst precursor was also reported for sulfated $ZrO_2/TiO_2$ nanocomposite synthesized by a hydrothermal synthesis approach [65]. However, the stable catalysts for esterification were reported for $WO_3/SiAl$ [43] and $ZrO_2/SBA-15$ [66] which were prepared by precipitation and impregnation processes,

respectively. Hence, suitable synthesis method for W-KIT-5 and W/A150 should be further investigated to improve the stability of these catalysts.

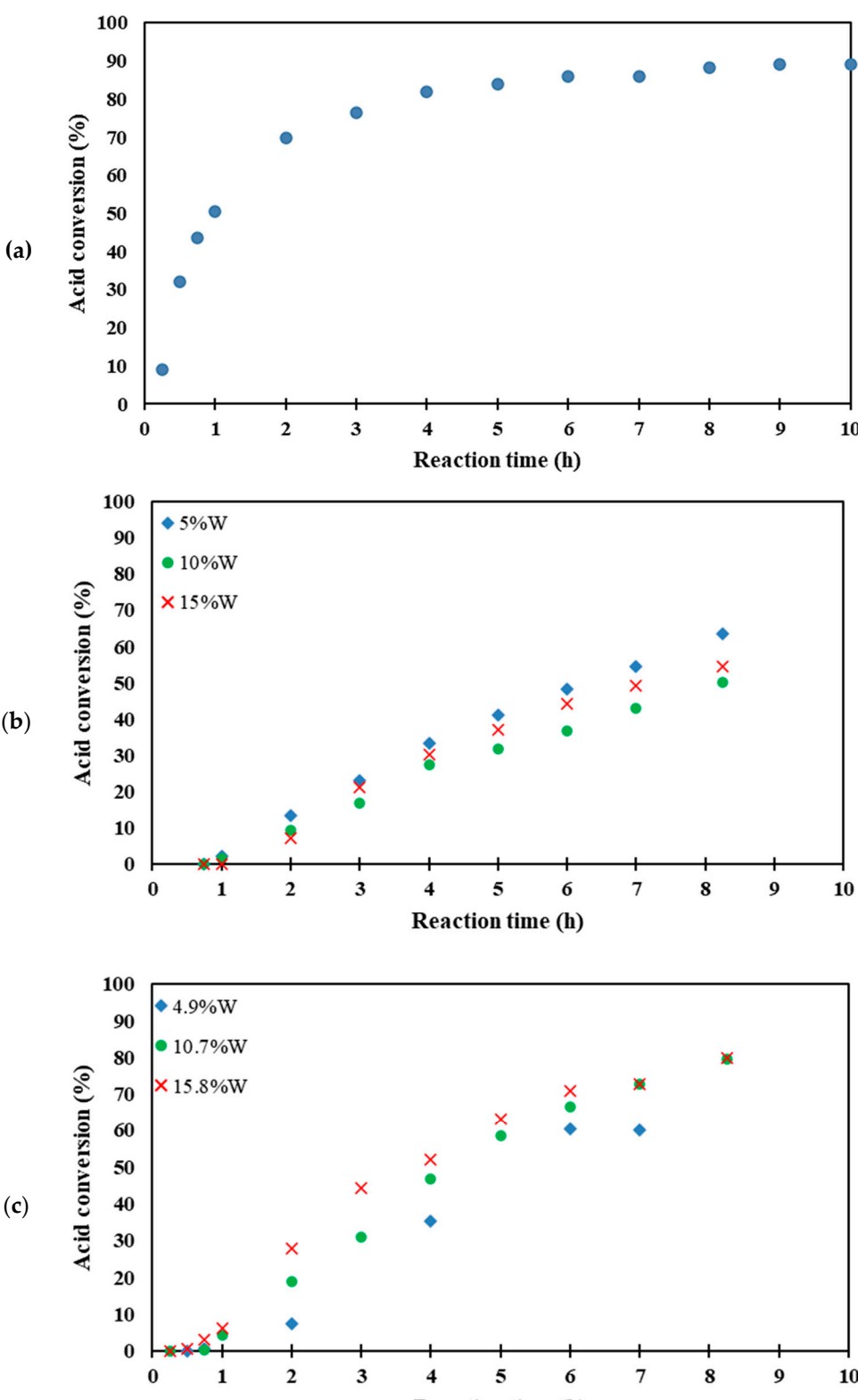

**Figure 4.** The esterification reaction rate with (**a**) tungstosilicic acid, (**b**) W/A150, and (**c**) W-KIT-5.

**Table 3.** Catalytic esterification of the bio-oil model.

| Catalyst (wt% of W) | Acid Conversion (%) |
|---|---|
| Without catalyst | 8 |
| Tungstosilicic acid | 87 |
| W/A150 (5%) | 59 |
| W/A150 (10%) | 52 |
| W/A150 (15%) | 50 |
| W-KIT-5 (4.9%) | 76 |
| W-KIT-5 (10.7%) | 81 |
| W-KIT-5 (15.8%) | 80 |

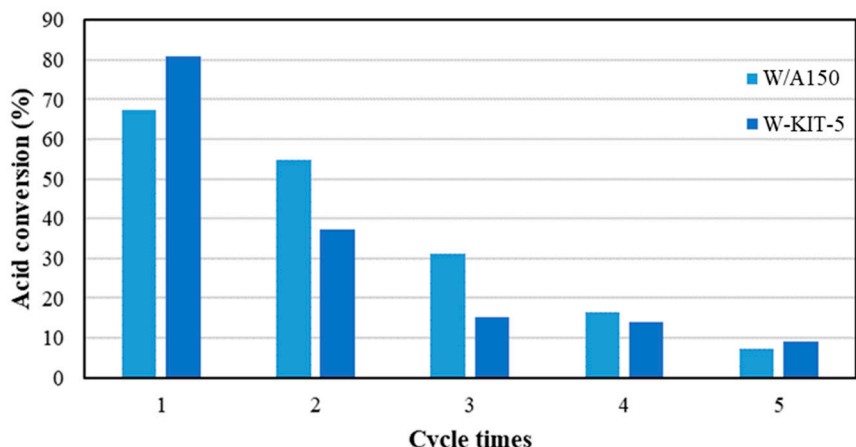

**Figure 5.** Reusability of the catalyst.

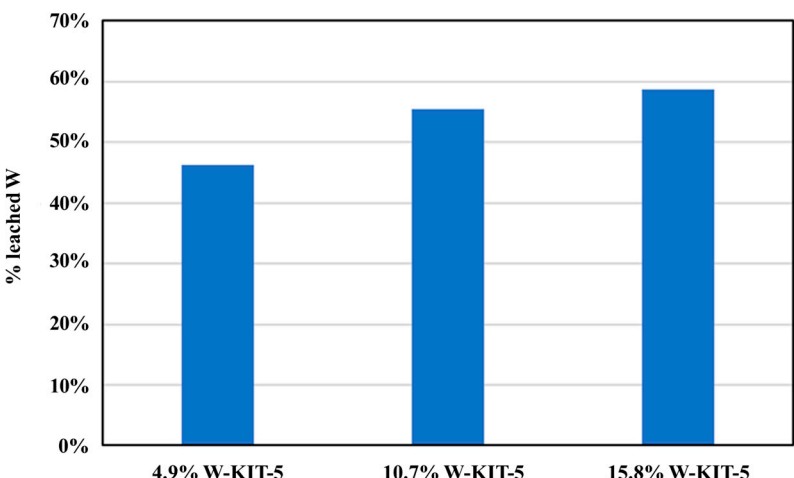

**Figure 6.** ICP analysis results of the W-KIT-5 catalysts after one use.

## 3. Materials and Methods

### 3.1. Materials

Ammonium metatungstate hydrate was purchased from Sigma-Aldrich (St. Louis, MI, USA). The amorphous silica powder was supplied by Evonik (Aerosil 150 or A150). Acetic acid (99% purity), ethanol (99.5% purity), and dioxane (99.8% purity) were purchased from Alpha Aesar (Haverhill, MA, USA), EMD Millipore (Burlingtonm, MA, USA), and Sigma Aldrich, respectively. WO$_3$ (99.995% trace metal basis) was purchased from Sigma Aldrich.

### 3.2. Catalyst Preparation

W/A150 was synthesized by incipient wetness impregnation. The required amount of ammonium metatungsate hydrate as the catalyst precursor was dissolved in 4 mL of deionized water and the mixture was well stirred. Then, 3 g of amorphous silica powder was added gradually to the solution to obtain a homogeneous paste. The paste was dried overnight on a vacuum filter at room temperature and then ground with a mortar and pestle. The impregnated samples were calcined at 300 °C for 2 h. Reference XRD samples for the W/A150 samples were prepared by physically mixing (via mortar and pestle) Aerosil 150 and crystalline $WO_3$ ($WO_3$ + A150) in calculated ratios to give W loadings equal to the W/A150 samples. KIT-5 and W-KIT-5 were provided by the University of Kansas. KIT-5 was prepared according to the procedure for high quality siliceous KIT-5 material synthesis reported by Kleitz et al. [50]. W-KIT-5 was prepared using a one-step hydrothermal synthesis procedure for synthesizing siliceous KIT-5 material reported by Ramanathan et al. [47]. The only difference was that the tungsten was sourced from ammonia metatungsate (Sigma Aldrich). The W/A150 catalyst system was synthesized with three different W concentrations: namely 5, 10, and 15 wt%. W-KIT-5 tungsten loadings were 4.9%, 10.7%, and 15.8%.

### 3.3. Catalyst Characterization

#### 3.3.1. X-ray Diffraction

X-ray diffraction (XRD) was used to investigate the crystalline size and structure of the samples. The XRD diffractograms were obtained using a Rigaku Miniflex II Powder X-ray diffractometer with a high sensitivity D/tex Ultra detector operating at 30 kV and 15 mA. The patterns were recorded in the angular range of 2θ = 10–80° using a 0.05° sampling interval and a counting time of 2.0° per min.

#### 3.3.2. X-ray Photoelectron Spectroscopy

X-ray photoelectron spectroscopy (XPS) analyses were performed using a Kratos AXIS Ultra DLD XPS system, with a monochromatic Al Kα source, operated at 15 keV and 150 W and a hemispherical energy analyzer. The X-rays were incident at an angle of 45° with respect to the surface normal and analysis was performed at a pressure below $1 \times 10^{-9}$ mbar. High resolution core level spectra were measured with a pass energy of 40 eV and survey scans with a pass energy of 160 eV. The analyses of the XPS spectra were performed with XPSPEAK 4.1 software. Each dataset was shifted according to the C-C bond known to exist at 284.6 eV.

#### 3.3.3. Brunauer–Emmett–Teller and Barrett–Joyner–Halenda Analyses

For physisorption characterization, the catalysts' surface area and pore size distribution were measured with the Brunauer–Emmett–Teller (BET) method using a Micromeritics ASAP 2020 instrument. The catalysts were degassed at 150 °C for 180 min, then analyzed with nitrogen adsorption and desorption isotherms. The desorption branch of the isotherms with the Barrett–Joyner–Halenda (BJH) method was used for the pore size distribution calculation.

#### 3.3.4. Temperature-Programmed Desorption of Ammonia Analysis

The acidity of the catalysts was performed by temperature-programmed desorption of ammonia ($NH_3$-TPD) method using a custom TPx with Inficon Transpector 2 Mass Spectrometer. Approximately 0.1 g of sample was pretreated in He for 1 h at 200 °C. The sample was then saturated with ammonia by flowing 2% $NH_3$/He at room temperature for about 40 min. The sample was further treated in pure He flows for about 30 min at room temperature to remove any physisorbed ammonia, before ramping at 10 °C/min in pure He flows to 600 °C. All total flow rates were 58 mL/min.

### 3.3.5. Inductively Coupled Plasma Analysis

For inductively coupled plasma (ICP) analysis, the collected reaction mixtures were heated in an oil bath at 100 °C to evaporate the organic liquid. A 10 mL of sample was mixed with 6 mL of 14.8 M lab grade $NH_4OH$ by shaking, then heated to a boil to dissolve any $WO_3$. The samples were then filtered with a 0.45 µm filter paper and diluted 10 times with deionized water before analysis in the ICP (Perkin-Elmer AVIO 200 ICP-OES).

### 3.4. Activity Evaluation of the Catalysts

The activity of W/A150, W-KIT-5, and tungstosilicic acid was evaluated for the esterification of acetic acid with ethanol. Typically, the reactions were carried out in a three-necked flask equipped with a thermometer, reflux condenser, and magnetic stir bar. The reaction mixture of acetic acid and ethanol with a molar ratio of 1:5 and dioxane as the internal standard (1000 µL: 1 g of acid) was mixed in the flask and then heated in a constant temperature bath [40]. The magnetic stirring rate was set at 500 rpm. A measure of 3 g of W catalyst per 100 g of acetic acid was added to the flask. The start time was set when the system reached the reaction temperature of 77 °C. The reaction solution was sampled from the flask every 15 min from start up to 1 h, and then every 60 min from 1 to 8 h. The samples were filtered to remove the catalyst and the composition was analyzed with an HP 5890 series II gas chromatograph. A Stabilwax column (30 m × 0.25 mm × 0.25 mm film thickness) was used to carry out the chromatographic separation. The injector and detector temperatures were set at 250 and 270 °C, respectively. The column temperature was programmed from 40 °C (held for 5 min) to 280 °C (held for 10 min) at a heating rate of 10 °C/min. The measurement of acetic acid was based on the internal standard of dioxane and reported as the average value of four independent replicates [67]. The conversion of acetic acid to ethyl acetate was calculated from the following formula: conversion = $(1 - AC_f/AC_p)$, in which $AC_f$ and $AC_p$ are the acid content of the feed and the final products, respectively.

### 3.5. Experiment of Model Bio-Oil Upgrading by Catalytic Esterification

Bio-oil upgrading with esterification process with W/A150, W-KIT-5, and tungstosilicic acid was studied and compared. In this work, acetic acid was used as the bio-oil model compound for study upgrading. The esterification was carried out in the three-necked flask equipped with a thermometer, a reflux condenser, and a magnetic stirrer heated in a constant temperature bath. In a typical reaction, the model reaction with acetic acid and ethanol by molar ratio of 1:5 was mixed in the flask. The catalyst used by 3 g of W per 100 g of acetic acid. The experiments were performed at the temperature of 77 °C for 8 h and 15 min with a stirring speed of 500 rpm [40]. After that, the catalyst was filtered and removed from the reacting mixture. The composition of both feed and product reaction mixtures and acid conversion were determined by the method previously proposed in this paper.

### 3.6. Evaluation of Catalyst Reusability

The reusability of W/A150 and W-KIT-5 catalysts was evaluated in sequential batch runs under the same operating condition with the esterification of bio-oil model compound as mentioned above. After each run, the catalyst was removed from the reaction mixture by filtration and washed with ethyl acetate, dried overnight on a vacuum filter at room temperature, and then ground with a mortar and pestle before adding it to the next reaction mixture. Five cycles were tested.

## 4. Conclusions

Tungsten oxides supported on amorphous silica (W/A150) and structured silica (W-KIT-5) were prepared by incipient wetness impregnation and a one-step hydrothermal method, respectively. All the synthesized catalysts had crystalline sizes less than 2.0 nm. The W/A150 catalysts were the best fit, with a hexagonal form of $WO_3$. XPS analysis

suggested the predominance of W 6+ species on higher loading W/A150 and W-KIT-5 catalysts. The W-KIT-5 catalysts appeared to have higher surface area and acidity than W/A150. The synthesized catalysts exhibited catalytic activity for the esterification of model bio-oil. The acid value of the bio-oil model compound was reduced by 59% and 81% using the 5% W/A150 and 10% W-KIT-5, respectively. This acid value reduction in W-KIT-5 was 93% of the homogeneous catalyst which demonstrated exceptional heterogeneous catalyst for bio-oil model compound upgrading via esterification. However, the significant leaching of tungsten from the support is the main issue that needs further improvement.

**Supplementary Materials:** The following supporting information can be downloaded at: https://www.mdpi.com/article/10.3390/catal13010038/s1, Figure S1: 15%, 10%, and 5% W/A150 XRD patterns showing for each sample the raw XRD pattern (W/A150), the raw reference pattern (WO$_3$ + A150) showing sharp peaks, and the background (sample holder plus A150 support) which was refined from the reference pattern and then used as the background for the analysis of the W/A150 samples.; Figure S2: W/A150 diffraction patterns after subtracting the background and silica support signals; Figure S3:15% W/A150: fit to hexagonal WO$_3$ (ICSD 32001); Figure S4: 10% W/A150: fit to hexagonal WO$_3$ (ICSD 32001); Figure S5: 5% W/A150: fit to hexagonal WO$_3$ (ICSD 32001); Table S1: Crystal Structures used to fit the isolated crystallite XRD signals for the WO$_3$/A150 samples. Quality of fit ($R_{wp}$) values are given for each individual structure's best fit to each of the W/A150 samples (smaller $R_{wp}$ = better fit) [47,52].

**Author Contributions:** Conceptualization, P.P., J.L. and J.R.R.; methodology, P.P. and J.L.; formal analysis, P.P., J.L. and A.D.; investigation, P.P., J.L. and A.D.; writing—original draft preparation, P.P. and J.L.; writing—review and editing, J.R.R. and N.T.; supervision, J.R.R. and N.T. All authors have read and agreed to the published version of the manuscript.

**Funding:** This research was partially funded by Thailand Research Fund, the Royal Golden Jubilee program, grant no. PHD 560651033, National Research Council of Thailand and Chiang Mai University.

**Data Availability Statement:** All relevant data are included in the paper and supplementary materials.

**Acknowledgments:** The authors were grateful for the kind support from Rajamangala University of Technology Thanyaburi, the University of South Carolina's SmartState Center of Catalysis for Renewable Fuels (CReF) and Catalysis for Renewables: Applications, Fundamentals and Technologies (CRAFT) collaboration. We also gratefully acknowledge John Monnier for reactor consulting, Stavros Karakalos for running the XPS and mentoring the XPS analysis, and Frank Girgsdies for mentoring the X-ray diffraction analysis.

**Conflicts of Interest:** The authors declare no conflict of interest.

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
