# Peer review of "Heterogenization of a Tungstosilicic Acid Catalyst for Esterification of Bio-Oil Model Compound"

_catalysts, doi:10.3390/catal13010038_

Round 1

Reviewer 1 Report

Tungston-containing solid acid catalyst for esterification is studied in present submission and suitable heterogenization way is displayed. The experiment phenomena are explained reasonably by means of characterizations

Questions

1 In Introduction, showing the objective reaction is encouraged.

2 Line 17, NH3 needs subscript.

3 Lines 172-173, "acetic acid and ethanol by molar ratio of 5:1" is miswritten.

4  Lines 225-230, for the acidity analysis in Table, illustrating the acidic centers'stucture for W/A150, W-KIT-5 and tungstosilicic acid is encouraged. 

Author Response

Comments

Reply

1. In Introduction, showing the objective reaction is encouraged.

Thank you for the recommendation. The introduction part was revised, showing the objective reaction. Lines 99-104

2. Line 17, NH3 needs subscript.

Revised, as suggested.

3. Lines 172-173, "acetic acid and ethanol by molar ratio of 5:1" is miswritten.

Revised, as suggested.

4. Lines 225-230, for the acidity analysis in Table, illustrating the acidic centers'stucture for W/A150, W-KIT-5 and tungstosilicic acid is encouraged.

Thank for the suggestion. Unfortunately, the acidic centers’ structures of W/A150, W-KIT-5 and tungstosilicic acid are not done in this work.

Reviewer 2 Report

The authors report on the attempt to turn from homogeneous to heterogeneous WOx-based acidic catalyst for bio-oil transesterification. They have synthesised two series of catalysts based on tungsten oxides using ammonim metatungstate as W-containing precursor and two types of SiO2-based materials as supports. Although the catalysts appeard to be not quite stable to the leaching in the reaction conditions, the obtained results are important for researchers working in this field. The paper can be published after addressing the following points:
- from the title it follows that the catalysts being developed are for real bio-oil transesterification while real bio-oil has not been tested, only mere model compounds (acetic acid, ethanol) were used. The crude bio-oil is known to be very complex feedstock for processig by any techniques. I would recommend to carry out a test the best supported catalysts for the transesterification of acids and alcohols in real bio-oil;
- values of acetic acid conversion in the presence of supported catalysts (Fig. 4b and 4c) in many cases are very close. What is the measurement error? It is reasonable to reflect it on the graphs. The same concerns such data in Table 2.

Author Response

Comments

Reply

1. From the title it follows that the catalysts being developed are for real bio-oil transesterification while real bio-oil has not been tested, only mere model compounds (acetic acid, ethanol) were used. The crude bio-oil is known to be very complex feedstock for processing by any techniques. I would recommend to carry out a test the best supported catalysts for the transesterification of acids and alcohols in real bio-oil

Thank you for your recommendation.

In this work, model bio-oil was utilized for the investigation catalytic acidity of the synthesized catalyst. Hence, the title of this work was revised to “…esterification of bio-oil model compound” to be in line with the work done.

The authors agree that bio-oil is interesting, then, the esterification of real bio-oil should be investigated further in the future study

2. Values of acetic acid conversion in the presence of supported catalysts (Fig. 4b and 4c) in many cases are very close. What is the measurement error? It is reasonable to reflect it on the graphs. The same concerns such data in Table 2.

Similar acetic acid conversion in different catalysts can happen which is reported in many research. However, the reaction rates obtained from the graph were determined by the slope of the graph as mentioned in section 3.2 which included several data points and can provide credible results. The acid conversion was calculated from the following formula: conversion = (1-ACf/ACp), in which ACf and ACp are the acid content of the feed and the final products, whose values were analyzed via a HP 5890 series II gas chromatography mentioned in lines 287 - 296. Therefore, the measurement error of resultant acid conversion is very small.

Reviewer 3 Report

Title: Heterogenization of a tungstosilicic acid catalyst for bio-oil esterification

Summary: Based on a prior demonstration of the high activity of a homogeneous tungstosilicic acid catalyst for the esterification of acetic acid, further study has been undertaken to heterogenize the catalyst. Significant leaching of tungsten from both supports occurred and will have to be solved in the future. The article is written well and supported by enough reference evidence and the following are further suggestions in improvements of the article.

Comments:

1.       Title: Can be more specific in accordance with the work done.

2.       Abstract: Need to be improved with obtained results.

3.       Keywords: Acceptable.

4.       Introduction: This section needs s to be elaborated with sufficient background research.

5.       The exploitation of cost-effective renewable heterogeneous base catalyst from banana (Musa paradisiaca) peel for effective methyl ester production from soybean oil can be useful in the improvement of the introduction.

6.       Previously published studies on enhanced biodiesel production via esterification of palm fatty acid distillate (PFAD) by using rice husk ash (NiSO4)/SiO2 catalyst could be supporting the elaboration on introduction.

7.       The novelty of the work needs to be specified and the objectives need to be more clarified.

8.       Material and methods: Need to be reproducible. Provide the refence for method applied, content wise acceptable.

9.       Catalyst characterization can be sub divided into different techniques applied.

10.   Results and discussion: The Results section is acceptable, but the discussion needs to be improved and elaborated on with the help of previously published articles.

11.   Section 3.3, 3.4 are missing comprehensive discussion and comparison with available literature. Excellent data is provided but authors need to be more focused on discussion.

12.   I suggest authors add comparative analysis table for the catalysts.

13.   Conclusion: This section needs improvements.

Minor Comments:

1.       High quality figures and graphs need to be included.

2.       The English language needs to be checked. Proofread can be done by native English speaker.

3.       Units need to be standardized.

4.       Grammatical mistakes need to be checked.

Author Response

Comments

Reply

1. Title: Can be more specific in accordance with the work done.

Thank you for your recommendation. Title revised, as suggested.

2. Abstract: Need to be improved with obtained results.

Revised, as suggested. Lines 18-24.

3. Keywords: Acceptable.

Thank you.

4. Introduction: This section needs to be elaborated with sufficient background research.

The introduction section elaborated with sufficient background, as suggested. Lines 73-98.

5. The exploitation of cost-effective renewable heterogeneous base catalyst from banana (Musa paradisiaca) peel for effective methyl ester production from soybean oil can be useful in the improvement of the introduction.

This research was referred to in this work, Line 73.

6. Previously published studies on enhanced biodiesel production via esterification of palm fatty acid distillate (PFAD) by using rice husk ash (NiSO4)/SiO2 catalyst could be supporting the elaboration on introduction.

This research was referred to in this work, Line 73-76.

7. The novelty of the work needs to be specified and the objectives need to be more clarified.

Revised, as suggested. Line 99-104.

8. Material and methods: Need to be reproducible. Provide the refence for method applied, content wise acceptable.

The detail about the catalyst preparation method and references were added, Lines 225-240.

9. Catalyst characterization can be sub divided into different techniques applied.

Revised, as suggested. Lines 214-277.

10. Results and discussion: The Results section is acceptable, but the discussion needs to be improved and elaborated on with the help of previously published articles.

Revised, as suggested. Lines 182-212.

11. Section 3.3, 3.4 are missing comprehensive discussion and comparison with available literature. Excellent data is provided but authors need to be more focused on discussion.

Revised, as suggested, Lines 182-212.

12. I suggest authors add comparative analysis table for the catalysts.

The reaction rate of all catalysts was summarized and added in Table 2, Lines 181.

13. Conclusion: This section needs improvements.

The conclusion section was revised to clearer content. Lines 317-330.

Minor Comments:

1. High quality figures and graphs need to be included.

All figures and graphs used are of high quality.

2. The English language needs to be checked. Proofread can be done by native English speaker.

The English language was checked and proofread by a co-author who is a native English speaker.

3. Units need to be standardized.

The standard unit was used in this work

4. Grammatical mistakes need to be checked.

Grammatical mistakes were checked again.

Round 2

Reviewer 2 Report

The article can now be published

Reviewer 3 Report

The authors provided results and discussion together and before the methods section. Anyway please check with MDPI guidelines. MDPI following following pattern such as

Abstract

Introduction

Material and methods

Results

Discussion

Conclusion